# Can You Trust Your Model? Constructing Uncertainty Approximations Guaranteeing Validity of Glioma Segmentation Explanations

**Tianyi Ren**[*1] [iD]                                                              TR1@UW.EDU
**Daniel Low**[*2]                                                               DALOW@UW.EDU
**Rachel Xiang**[†3]                              RACHEL.XIANG@PENNMEDICINE.UPENN.EDU
**Pittra Jaengprajak**[†2]                                                  JPITTRA@UW.EDU
**Juampablo Heras Rivera**[1]                                                  JEHR@UW.EDU
**Riley Olson**[1]                                       RILEY.HANNAH.OLSON@GMAIL.COM
**Jacob Ruzevick**[4]                                                     RUZEVICK@UW.EDU
**Mehmet Kurt**[1]                                                         MKURT@UW.EDU

[1] *Department of Mechanical Engineering, University of Washington, Seattle, USA*

[2] *University of Washington School of Medicine, Seattle, USA*

[3] *Perelman School of Medicine, University of Pennsylvania, Philadelphia, USA*

[4] *Department of Neurological Surgery, University of Washington, Seattle, USA*

**Editors:** Accepted for publication at MIDL 2026

## Abstract

Deep learning models have been successfully applied to glioma segmentation from multi-contrast MRI, yet model reasoning is difficult to validate clinically. Prior work used contrast-level Shapley values to explain how individual MRI sequences contribute to segmentation performance, and showed that alignment between these explanations and protocol-derived contrast rankings is associated with improved model performance. However, a single trained model may not reflect the optimal population-level model, and naive Deep Ensemble uncertainty estimates provide no guarantees that the true optimal explanation lies within their intervals. In this work, we construct statistically valid uncertainty intervals for contrast-level Shapley values in glioma segmentation. Using a U-Net trained on the BraTS 2024 GoAT dataset, we compute Shapley values for each MRI contrast and tumor sub-region, form naive uncertainty estimations from cross-validation, and then apply a frequentist framework based on uniform convergence to define a confidence set of plausibly optimal models. By optimizing mixed objectives that trade off empirical loss and Shapley value, we approximate the Pareto frontier and obtain lower and upper bounds on the optimal explanation. We compare these intervals with clinically derived consensus and protocol rankings. Our results demonstrate that naive uncertainty estimations can lead to inconclusive or misleading conclusions about clinical alignment, whereas frequentist intervals provide principled guarantees on coverage of the optimal explanation and show moderate correlation with annotator consensus, enabling more reliable validation of model explanations against established clinical reasoning.

**Keywords:** Uncertainty, Explainable AI, Deep Learning, MRI

---

[*] Contributed equally

[†] Contributed equally

## 1. Introduction

Automated segmentation of brain tumors is an essential task for computer-aided diagnosis, guiding treatment, planning, and monitoring of gliomas (Hesamian et al., 2019). Deep learning models now achieve impressive performance on this task, yet their "black box" nature limits clinical applications (Rudin, 2019). Medical decision-making requires transparency, as erroneous model reasoning can lead to missed tumor regions and adverse patient outcomes. (Liu et al., 2019). Thus, medical segmentation models require not only high accuracy but also clinically interpretable explanations that enable model comparison and validation for clinical use.

We previously used Shapley values to quantify the contribution of each MRI contrast to glioma segmentation. This approach assigns importance at the contrast level by systematically perturbing model inputs and measuring the impact on segmentation performance (Ren et al., 2025b). Alternative explainability methods, such as Grad-CAM, generate pixel level feature importance maps highlighting which spatial regions influence predictions (Selvaraju et al., 2017). However, they require ad-hoc thresholding to delineate important regions and have been shown to lack reproducibility and localization accuracy in medical imaging (Venkatesh et al., 2024). Clinicians do not necessarily reason in terms of pixel-level importance; rather, established protocols for glioma segmentation specify which MRI contrasts to prioritize for identifying specific tumor sub-regions (e.g., T1c for enhancing tumor, T2-FLAIR for peritumoral edema) (Ellingson et al., 2015; Baid et al., 2021). Contrast-level Shapley values directly parallel this clinical reasoning, enabling comparison between model explanations and protocol-based contrast rankings. In our previous work, we found that model performance correlated significantly with alignment between Shapley-derived rankings and clinical protocol rankings, suggesting that *clinically-aligned feature usage may characterize optimal segmentation models* (Ren et al., 2025a).

While these results highlight the utility of Shapley values for clinically oriented explanations, a single model's explanation does not necessarily reflect the "ground truth" feature attribution. This is due to the phenomenon of model multiplicity, where distinct models with comparable performance exhibit different explanations, meaning any individual trained model may have idiosyncratic feature dependencies (Breiman, 2001). In glioma segmentation, a model may achieve deceptively high Dice scores while systematically under-

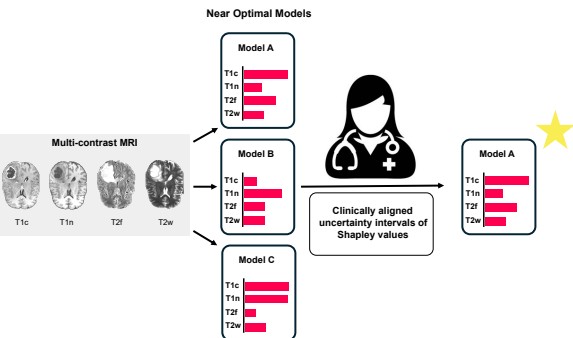

Figure 1: Overview of uncertainty intervals used for model selection.

leveraging T2-FLAIR, thereby failing to delineate the infiltrating tumor margins critical for treatment planning. Explaining one of the best-performing models cannot determine whether its feature usage represents the optimal, population-level explanation (Fisher et al., 2019). Moreover, naive Monte Carlo uncertainty estimates derived from cross-validation folds or model ensembles lack the necessary theoretical guarantees to determine if clinical reasoning accurately approximates optimal model behavior. Therefore, principled uncertainty quantification (UQ) methods are essential. These methods can construct provably reliable uncertainty intervals that are guaranteed to contain the optimal model explanation, thereby enabling rigorous validation of model explanations against established clinical protocols (Fig. 1) (Marx et al., 2023).

In this study, we apply a frequentist uncertainty quantification framework for image classification to construct intervals for contrast-level Shapley values for multi-contrast image segmentation. Rather than explaining a single trained model, we target the "optimal model", the model that would minimize loss over the entire patient population, representing the true relationship between MRI contrasts and glioma segmentation (Marx et al., 2023). The resulting uncertainty intervals are mathematically guaranteed to contain the optimal model's explanation with high probability. Our results demonstrate that naive Monte Carlo uncertainty estimations can lead to inconclusive or misleading conclusions about clinical alignment because the explanation interval does not cover the optimal model's explanation, whereas frequentist intervals provide principled guarantees on coverage of the optimal explanation and show moderate correlation with annotator consensus. Our contributions are as follows:

- We developed an uncertainty quantification framework to establish statistically valid confidence intervals for contrast-level Shapley values for glioma segmentation models. These Shapley value intervals are guaranteed to encompass the true, optimal model explanation.

- We compare naive uncertainty estimates from cross-validation with frequentist intervals, demonstrating the limitations of naive Monte Carlo approaches for explanation interval estimation.

- We evaluate whether clinical protocol rankings are covered by valid uncertainty intervals, testing the alignment between clinical reasoning and optimal model behavior.

## 2. Methods

### 2.1. Dataset and Model Architecture

We evaluated glioma segmentation using the BraTS Challenge 2024 GoAT dataset, comprising 1,351 patients. Each case included four co-registered MRI sequences: contrast-enhanced T1-weighted (T1c), native T1-weighted (T1n), T2-weighted fluid-attenuated inversion recovery (T2-FLAIR), and T2-weighted (T2w) imaging. Expert annotations delineated three tumor sub-regions: enhancing tumor, necrotic core, and peritumoral edema, and segmentation accuracy was quantified using the Dice similarity coefficient (de Verdier et al., 2024).

We define the multi-contrast segmentation problem as follows: Let $N$ denote the set of MRI contrasts. Given a multi-channel input $I \in \mathbb{R}^{|N| \times H \times W \times D}$, the deep learning model $\omega$ is trained to predict the tumor segmentation $\omega(I)$.

A standard multi-channel 3D U-Net and the implementation provided by the MONAI library (Cardoso et al., 2022) with four encoder–decoder stages and skip connections, taking four MRI contrasts as input and output three channel output representing each tumor subregion. U-Net models were trained and evaluated using five-fold cross-validation with Dice loss and Focal Loss (Ren et al., 2025b, 2023, 2024). The combined loss function $\mathcal{L}$ used for training is defined as:

$$\mathcal{L} = \lambda_{\text{Dice}} \cdot \mathcal{L}_{\text{Dice}} + \lambda_{\text{Focal}} \cdot \mathcal{L}_{\text{Focal}} \tag{1}$$

where $\lambda_{\text{Dice}}$ and $\lambda_{\text{Focal}}$ are weighting coefficients, $\lambda_{\text{Dice}} + \lambda_{\text{Focal}} = 1$, $\lambda_{\text{Dice}}, \in [0, 1]$. The Dice loss is defined as $\mathcal{L}_{\text{Dice}} = 1 - \frac{2 \sum_i p_i g_i}{\sum_i p_i^2 + \sum_i g_i^2}$ for each tumor region individually, where $p_i$ and $g_i$ are the predicted and ground truth values for the $i-$th contrast. The index iterates over $i \in \{1, \cdots, |N|\}$, where the set $N = \{n_1, n_2, n_3, n_4\}$ represents the four MRI contrasts: T1c, T1n, T2-FLAIR, and T2w. The Focal loss is defined as $\mathcal{L}_{\text{Focal}} = -\alpha(1 - p_r)^\gamma \log(p_r)$, where $p_r$ is the predicted probability for the true class, $\alpha$ is a balancing factor, $\gamma$ is the focusing parameter, and $r \in \{1, 2, 3\}$ indexes the tumor sub-regions (enhancing tumor, necrotic core, edema).

## 2.2. Contrast Level Shapley Values

The contribution of each MRI contrast was quantified using the Shapley value framework (Ren et al., 2025b). Let $\omega$ denote a trained segmentation model. Given a multi-channel input $I \in \mathbb{R}^{4 \times H \times W \times L}$, the model predicts tumor segmentation $\omega(I)$.

The contrast-level Shapley value $\phi_i^{(r)}(\omega)$ quantifies the contribution of the $i-$th contrast $(n_i)$ to segmentation performance for tumor sub-region $r$ by averaging its marginal contribution across all possible subsets $S$ of contrasts, weighted by subset size:

$$\phi_i^{(r)}(\omega) = \sum_{S \subseteq N \setminus \{n_i\}} \frac{|S|!(|N| - |S| - 1)!}{|N|} \left( D_\omega^{(r)}(S \cup \{n_i\}) - D_\omega^{(r)}(S) \right) \tag{2}$$

where $D_\omega^{(r)}(S)$ denotes the Dice score achieved by model $\omega$ for sub-region when using only the subset $S \subseteq N$ of contrasts as input. We computed Shapley values $\phi_i^{(r)}(\omega)$ across all five cross-validation folds.

## 2.3. Contrast Level Clinical Rankings

Two clinical ranking systems were established to compare against model-derived explanations. Both were evaluated on a cohort of 202 patients selected from the BraTS 2024 GoAT dataset: all cases with mean Dice scores below 0.5 across folds ($n = 122$) plus a random sample of 80 cases with Dice scores above 0.5 (de Verdier et al., 2024). This cohort was intentionally weighed towards low scoring subjects to ensure subjective rankings (see below) remained stable even for "difficult" cases, as contrast rankings according to clinical protocol are globally enforced on all subjects (Baid et al., 2021).

For the *consensus ranking*, three U.S. medical students independently ranked the four MRI contrasts (1–4, with 1 indicating highest importance) for each case, following guidance from the BraTS 2021 Benchmark paper (Baid et al., 2021). Ties were permitted. The primary criterion was the number of tumor sub-regions identifiable in each contrast, with ground truth annotations used to verify presence or absence of sub-regions. To obtain dataset-level consensus values, all rankings were pooled across subjects and raters for each contrast, yielding a mean ranking with 95% confidence intervals calculated using the *t*-distribution.

For the *protocol ranking*, we applied fixed rankings derived from the BraTS 2021 paper: T1c = 1, T2-FLAIR = 2, and T1n/T2w = 3 (Baid et al., 2021). The consensus ranking captures inter-rater and inter-subject variability in clinical judgment, while the protocol ranking represents standardized clinical guidance. The consensus and protocol ranking will be collectively referred to as *clinical rankings*.

### 2.4. Naive Deep Ensemble Rank Interval

As previously expressed, uncertainty estimates from cross-validation folds are not statistical guarantees of optimal explanation. To illustrate this point, we employed a Deep Ensemble uncertainty interval to compare to clinical rankings. We first computed Shapley values $\phi_i^{(r,f,m)}(\omega)$ for each contrast $n_i$, across all tumor sub-region $r \in \{1,2,3\}$, cross-validation fold $f \in \{1, \cdots, 5\}$, and subject $m \in \{1, \cdots, 1351\}$ for the BraTS 2024 dataset. There is a set of Shapley values within each region-fold combination $(r, f)$ for all subjects $m$. We ranked the four contrasts based on their Shapley values in descending order, assigning rank 1 to the contrast with the highest Shapley value and rank 4 to the lowest. Let $R_i^{(r,f,m)}$ denote the rank of contrast $n_i$ for subject $m$ in region $r$ and fold $f$, where $R_i^{(r,f,m)} \in \{1,2,3,4\}$. To obtain naive Deep Ensemble uncertainty intervals, we pooled all ranking values across subjects, regions, and folds. For each contrast $n_i$, this yielded $M \times 3 \times 5 = 20{,}265$ ranking values (where $M = 1{,}351$ subjects). The dataset mean ranking was computed as:

$$\bar{R}_i^{\text{pop}} = \frac{1}{M \cdot 3 \cdot 5} \sum_{m=1}^{M} \sum_{r=1}^{3} \sum_{f=1}^{5} R_i^{(r,f,m)} \tag{3}$$

with 95% confidence intervals calculated for each contrast using the t-distribution. T-tests were then performed to directly compare these U-Net Shapley ranking naive Deep Ensemble uncertainty intervals with 95% confidence intervals of clinical rankings 2.3. The purpose of this comparison is to highlight the initial mismatch between a set of U-Net derived rankings and our clinical rankings and the need for a more rigorous statistical guarantee of the optimal model.

### 2.5. Uncertainty Intervals for Shapley Values through Conformal Analyisis

**The Optimal Model:** Standard explainability approaches compute $\phi_i(\omega)$ for a single trained model $\omega$ and treat these Shapley values as representative of model behavior. However, the phenomenon of model multiplicity, where multiple models achieve similar Dice scores but exhibit different feature usage patterns means a single model's Shapley values may not reflect the optimal model $\omega^*$ for the glioma population. Thus, we want to create

uncertainty intervals $C = C(B)$ for the optimal model using an empirical dataset $B$, which for our purposes is the BraTS Challenge 2024 GoAT dataset. This uncertainty interval contains the true explanation for the $i$-th contrast of the optimal model with high probability for a confidence level $1 - \alpha$:

$$P(\phi_i(\omega^*) \in C) \geq 1 - \alpha, \alpha \in (0, 1) \tag{4}$$

where $\phi_i(\omega) = \frac{1}{3}\sum_{r=1}^{3}\phi_i^{(r)}(\omega)$ denotes the Shapley value for the $i-$th contrast averaged across tumor sub-regions.

**Naive Deep Ensemble Comparison:** Naive estimations illustrate model multiplicity and do not have statistical guarantees of containing the optimal explanation (Marx et al., 2023). The U-Net Shapley values $\phi_i^{(r,f,m)}(\omega)$ (2.4) for each tumor sub-region, fold, and subject were plotted to create a spread of Shapley values with calculation of the mean for comparison to the 95% confidence intervals of each MRI contrast from the clinical ranking cohort (2.3). Because the clinical rankings cannot be directly compared to the U-Net Shapley value intervals, we employed Kendall's Tau to compare the *rank correlation*. Kendall's Tau measures alignment of the order of each MRI contrast between the U-Net Shapley value intervals (naive estimation) and the clinical ranking confidence intervals. We expect to see weak or insignificant alignment as we hypothesize the clinical rankings to approximate the *optimal explanation* that is not contained in the naive estimation.

Figure 2: Clinical rankings (1-4) cannot be directly compared to Shapley value intervals (unbounded). Thus, Shapley value intervals of each contrast are converted to rankings (1-4) based on interval mean and Kendall's Tau is used to compare rank correlation. This method is our best attempt to assess alignment between clinical rankings and Shapley value intervals.

**Frequentist Uncertainty Intervals:** Our approach relies on the assumption that the model class $\Omega$ satisfies uniform convergence, i.e., the empirical loss $L_{\text{emp}}(\omega)$ from the given dataset $B$ converges uniformly to the population loss $L(\omega)$ across all models $\omega$ in $\Omega$.

To quantify the uncertainty and stability of our Shapley value explanations, we adopt the methodology proposed by Marx et al. (Marx et al., 2023) to define a confidence set of plausibly optimal models. This approach leverages the concept of $(\alpha, \epsilon_n)$-uniform convergence, which ensures that the difference between the population loss $L(\omega)$ and the empirical loss $L_{\text{emp}}(\omega)$ is bounded by $\epsilon_n$ with probability $1 - \alpha$:

$$\sup_{\omega \in \Omega} |L(\omega) - L_{\text{emp}}(\omega)| \leq \epsilon_n. \tag{5}$$

Based on this guarantee, a confidence set of plausibly optimal models, $\Omega_\alpha$, is defined as those models whose empirical loss is bounded by $2\epsilon_n$ above the empirical risk minimizer $(\hat{\omega})$:

$$\Omega_\alpha = \{\omega \in \Omega : L_{\text{emp}}(\omega) \leq \inf_{\omega' \in \Omega} L_{\text{emp}}(\omega') + 2\epsilon_n\}. \tag{6}$$

Following Marx et al.'s approach, we approximate the bounds of the resulting explanation set $C_{freq}$ by tracing the Pareto frontier using a sequence of mixture weight optimizations, as shown in Equations (7) and (8). For a sequence of mixture weights $\lambda \in [0, 1]$, we optimize:

$$\hat{\omega}_\lambda^- = \arg\min_{\omega \in \Omega} \lambda\phi_i(\omega) + (1 - \lambda)L_{\text{emp}}(\omega) \tag{7}$$

$$\hat{\omega}_\lambda^+ = \arg\min_{\omega \in \Omega} -\lambda\phi_i(\omega) + (1 - \lambda)L_{\text{emp}}(\omega). \tag{8}$$

The lower and upper bounds of $C_{freq}$ are then:

$$\text{Lower: } \min\{\phi_i(\hat{\omega}_\lambda^-) : L_{\text{emp}}(\hat{\omega}_\lambda^-) \leq L_{\text{emp}}(\hat{\omega}) + 2\epsilon_n\}, \tag{9}$$

$$\text{Upper: } \max\{\phi_i(\hat{\omega}_\lambda^+) : L_{\text{emp}}(\hat{\omega}_\lambda^+) \leq L_{\text{emp}}(\hat{\omega}) + 2\epsilon_n\} \tag{10}$$

where $\hat{\omega}$ is the empirical risk minimizer. In our implementation, we set $\alpha = 0.05$ and $\epsilon_n = 0.05$. The $2\epsilon_n = 0.10$ tolerance allows models whose loss is at most 0.10 higher than the empirical risk minimizer. Our best model achieves 87% Dice, so we include models with a minimum Dice score of 77% (Table 1). A minimum Dice of 77% adequately captures general human-expert annotation accuracy as well as top performing tumor segmentation models available (Pemberton et al., 2023; Tillmanns et al., 2022; Akanuma et al., 2025). The complete procedure is detailed in Algorithm 1. If clinical rankings represent the explanation of the true population-level relationship between MRI contrasts and glioma segmentation, valid uncertainty intervals should align with these clinical values. The comparison is again assessed using Kendall's Tau, since the frequentist intervals cannot be directly compared to clinical rankings and rank correlation is used instead. Alignment of clinical rankings with $C_{freq}$ would support the hypothesis that established clinical reasoning reflects optimal feature usage, while exclusion would suggest either model misspecification or that clinical protocols do not approximate the true optimal explanation.

**Detailed Explanation of Algorithm 1:** Algorithm 1 outlines the procedure for constructing uncertainty intervals for Shapley values using a frequentist approach. The algorithm begins by estimating the Empirical Risk Minimizer (ERM) $\hat{\omega}$ and its corresponding empirical risk $\mathcal{L}_{\text{emp}}(\hat{\omega})$. It then iteratively optimizes a mixed objective function that balances the Shapley value $\phi(\omega)$ and the empirical loss $\mathcal{L}_{\text{emp}}(\omega)$ for a sequence of mixture

---

**Algorithm 1** Construction of Frequency-Based Confidence Intervals for Explanations

---

1: **Input:**
2:     Dataset $D$
3:     Sequence of mixture weights $\Lambda = \{\lambda_1, \ldots, \lambda_K\}$, where $0 \leq \lambda_1 < \cdots < \lambda_K \leq 1$
4:     Uniform convergence bound $\epsilon_n$
5: **Initialization:**
6: Train the ERM model $\hat{\omega}$ on $D$ (e.g., via $K$-fold cross-validation)
7: Compute the empirical risk $\mathcal{L}_{\text{emp}}(\hat{\omega})$
8: Initialize sets: $\mathcal{S}^- \leftarrow \emptyset$, $\mathcal{S}^+ \leftarrow \emptyset$
9: **Pareto Frontier Tracing:**
10: **for** $\lambda \in \Lambda$ **do**
11:     ▷ Compute model minimizing the explanation value (Lower Bound Trace)
12:     $\hat{\omega}_\lambda^- = \arg\min_{\omega \in \Omega} \ \lambda\phi(\omega) + (1-\lambda)\mathcal{L}_{\text{emp}}(\omega)$
13:     $\mathcal{S}^- \leftarrow \mathcal{S}^- \cup \{(\phi(\hat{\omega}_\lambda^-), \mathcal{L}_{\text{emp}}(\hat{\omega}_\lambda^-))\}$
14:     ▷ Compute model maximizing the explanation value (Upper Bound Trace)
15:     $\hat{\omega}_\lambda^+ = \arg\min_{\omega \in \Omega} \ -\lambda\phi(\omega) + (1-\lambda)\mathcal{L}_{\text{emp}}(\omega)$
16:     $\mathcal{S}^+ \leftarrow \mathcal{S}^+ \cup \{(\phi(\hat{\omega}_\lambda^+), \mathcal{L}_{\text{emp}}(\hat{\omega}_\lambda^+))\}$
17: **end for**
18: **Determine Confidence Bounds:**
19: $\tau = \mathcal{L}_{\text{emp}}(\hat{\omega}) + 2\epsilon_n$
20: $\hat{\phi}_{\text{lower}} = \min\{\phi : (\phi, L) \in \mathcal{S}^-, \ L \leq \tau\}$
21: $\hat{\phi}_{\text{upper}} = \max\{\phi : (\phi, L) \in \mathcal{S}^+, \ L \leq \tau\}$
22: **Return:** $\mathcal{C}_{\text{freq}} = [\hat{\phi}_{\text{lower}}, \hat{\phi}_{\text{upper}}]$

---

weights $\lambda \in [0, 1]$. By adjusting $\lambda$, the algorithm traces the Pareto frontier between explanation values and empirical loss. For each $\lambda$, two optimizations are performed: one to minimize the Shapley value and another to maximize it, while ensuring the empirical loss remains within a specified tolerance. The lower and upper bounds of the confidence interval are determined by selecting the first points on the Pareto frontier that satisfy the loss constraint. This approach provides an approximate but valid confidence interval for the optimal Shapley value, enabling rigorous validation of model explanations against clinical protocols. In our experiments, we selected $\lambda$ values uniformly from 0 to 1 with a step size of 0.2. For each MRI contrast, we performed separate optimization procedures to obtain the lower and upper explanation bounds, requiring two sets of experiments per contrast to estimate $\hat{\phi}_{\text{lower}}$ and $\hat{\phi}_{\text{upper}}$. All experiments were conducted using four NVIDIA A40 GPUs, with each model training requiring approximately 12 GPU-hours results in a total of 576 GPU-hours to get the Confidence Intervals.

## 3. Results

The results of our experiments are presented in two main sections: the initial Naive Deep Ensemble Rank Interval (2.4) and the Uncertainty Intervals for Shapley Values (2.5). We first train a five-fold cross validation to obtain the empirical loss and subjects' Shapley value. The five cross-validation folds of the U-Net model achieved mean Dice scores of 83.66%, 86.58%, 91.17% for necrotic core, enhancing tumor, and edema, respectively. The detailed results of which are shown in Table 1.

Table 1: Model performance metrics across folds (Baseline).

| Fold | Baseline | | | |
|---|---|---|---|---|
| | Dice Score [-]↑ | | | |
| | NCR | ET | ED | Average |
| Fold 0 | 84.61% | 86.73% | 91.10% | 87.48% |
| Fold 1 | 83.02% | 87.30% | 91.63% | 87.32% |
| Fold 2 | 83.45% | 86.27% | 91.32% | 87.01% |
| Fold 3 | 84.93% | 86.64% | 90.60% | 87.39% |
| Fold 4 | 82.31% | 85.97% | 91.18% | 86.49% |
| **Average** | 83.66% | 86.58% | 91.17% | 87.14% |

### 3.1. Naive Deep Ensemble Rank Interval

T-tests of mean rankings between U-Net Shapley rankings and consensus and protocol rankings for each MRI contrast are shown in Figure 3. Results demonstrate significant differences between U-Net Shapley rankings and both consensus and protocol rankings across all MRI contrasts.

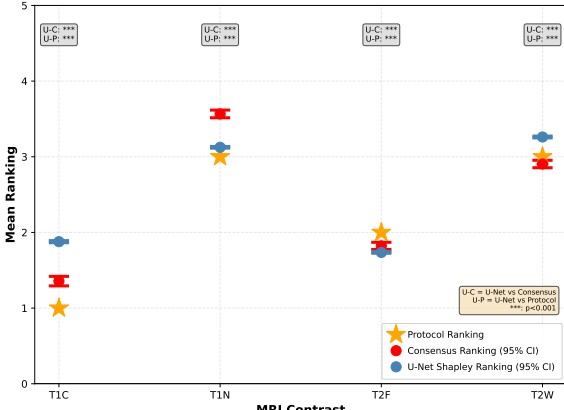

Figure 3: T-test comparison of means between U-Net Shapley Rankings and consensus and protocol rankings shows significant differences across all contrasts. Note: The figure displays the mean and 95% confidence intervals for the U-Net Shapley Ranking (U), Consensus Ranking (C), and Protocol Ranking (P) across four MRI contrasts, with annotated grey boxes showing the statistical $t$-test $p$-values for the paired comparisons U-C and U-P.

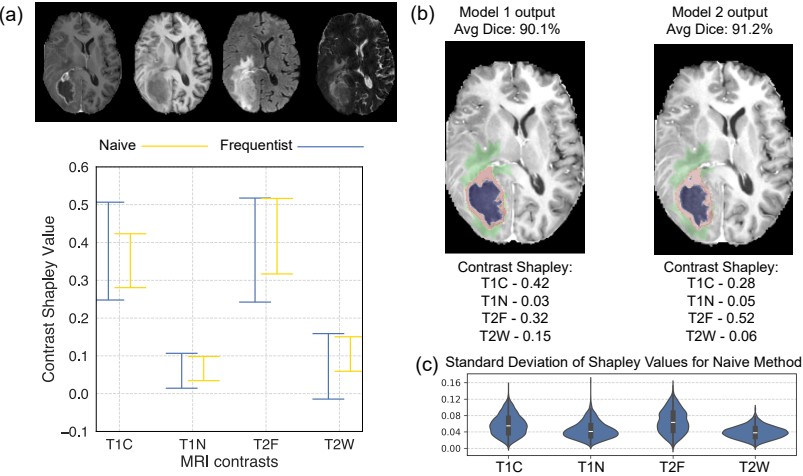

Figure 4: (a) Comparison of uncertainty intervals for Shapley values: the statistically valid frequentist interval (Blue) versus the naive interval (Yellow). (b) Visualization of multiplicity of model explaination, demonstrating that high-performing models can exhibit significant variation in contrast-level Shapley values. (c) Standard deviation of contrast Shapley values estimated using the naive approach.

Table 2: Rank correlation between Shapley value intervals (upper bound, lower bound, and mean) and both consensus and protocol rankings.

| Method | Bound | vs. Consensus | | vs. Protocol | |
|--------|-------|---------------|---------|---------------|---------|
| | | Kendall's $\tau$ | p-value | Kendall's $\tau$ | p-value |
| Naive Deep Ensemble | Upper | 0.206 | 0.350 | 0.125 | 0.540 |
| | Lower | 0.056 | 0.087 | 0.156 | 0.130 |
| | Mean | 0.158 | 0.426 | 0.207 | 0.071 |
| Frequentist | Upper | **0.467** | **0.015** | 0.256 | 0.214 |
| | Lower | 0.256 | 0.214 | 0.312 | 0.221 |
| | Mean | **0.371** | **0.042** | 0.401 | 0.089 |

### 3.2. Uncertainty Intervals for Shapley Values

Table 2 presents the Kendall's Tau ($\tau$) rank correlation analysis between the U-Net Shapley value distribution intervals (Upper, Lower, and Mean) and both consensus and protocol rankings. The Naive Deep Ensemble method generally showed no correlations with the consensus or the protocol ranking. In contrast, the Frequentist method demonstrated statistically significant moderate agreement with Consensus rankings for the Upper Bound ($\tau = 0.467$, $p = 0.015$) and the Mean ($\tau = 0.371$, $p = 0.042$). Correlations between the Frequentist method and Protocol rankings were not statistically significant ($p > 0.05$).

## 4. Discussion

In this work, we propose and validate a framework for calculating the optimal explanation interval for glioma segmentation explanations by explicitly targeting the explanation of the optimal population-level model rather than that of a single trained network. We compared naive uncertainty estimates derived from cross-validation against a frequentist framework based on uniform convergence. Our analysis revealed that naive intervals without statistical guarantees failed to align with our clinical rankings. Importantly, the frequentist intervals show moderate correlation with the consensus rankings. Because the frequentist intervals guarantee with high probability the containment of the optimal explanation of the population, the results suggest that human-like reasoning found in the consensus protocol converges toward the optimal explanation.

The divergence between Protocol and Consensus rankings (Table 2) reflects an interesting finding in this study. First, our consensus cohort was strategically designed to include 122 challenging cases with Dice scores below 0.5, representing diagnostically difficult tumors where standard protocol guidance may be insufficient. In these cases with ambiguous tumor boundaries, atypical presentations, or extensive infiltration, rigid protocol rankings (T1c=1, T2-FLAIR=2, T1n/T2w=3) cannot capture the case-specific variability that human annotators naturally incorporate when prioritizing contrasts. When asked for feedback on the ranking process, medical student annotators reported cases where one of T1n/T2w were severely worse than the other, such that the tumor could not be visualized at all. While this detracts from the clinical protocol established from Baid et al., it highlights important

situational decision making that may better approach the optimal explanation for glioma segmentation.

This convergence between clinical reasoning and optimal model behavior has practical implications for the persistent challenge of model multiplicity (Figure 4 b & c). When many models achieve comparable performance yet differ in feature importance, selecting a model for clinical use is difficult. Choosing a model that achieves high Dice scores while under-leveraging clinically critical contrasts(such as T2-FLAIR for infiltrating tumor margins) may produce segmentations that fail, impacting tumor treatment planning. By validating consensus-derived rankings with frequentist intervals, we establish a principled criterion for selection: distinguishing models that align with radiological protocols from those that, despite high predictive accuracy, lack clinical grounding.

Several directions remain for future investigation. First, validating this framework across diverse architectures, such as transformer-based models, would establish whether the alignment between consensus rankings and optimal explanations generalizes beyond convolutional networks. Second, while our consensus rankings were derived from medical students following established BraTS guidelines, obtaining rankings from neuroradiologists would provide a more authoritative clinical benchmark for selecting clinically aligned glioma segmentation models.

## 5. Conclusion

By constructing statistically valid uncertainty intervals for the optimal model's Shapley values, our frequentist framework provides a principled and trustworthy methodology for rigorously validating whether established clinical protocols accurately reflect the optimal feature usage in deep learning-based glioma segmentation.

## Acknowledgments

This research is funded by NSF Grant No.CMMI-1953323.

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

## Appendix A.

We conduct a sensitivity analysis on the uniform convergence tolerance $\epsilon_n$ used to define the confidence set of plausibly optimal models. In the main experiments, we set $\epsilon_n = 0.05$. Here, we increase the tolerance to $\epsilon_n = 0.1$, corresponding to a looser loss bound for a larger set of optimal models. The results show that there is no correlation between the Shapley interval rankings and either the consensus or the protocol rankings (Table S1).

Table S1: Rank correlation between Shapley value intervals (upper bound, lower bound, and mean) and both consensus and protocol rankings when $\epsilon_n = 0.1$.

| Method | Bound | vs. Consensus | | vs. Protocol | |
|---|---|---|---|---|---|
| | | Kendall's $\tau$ | p-value | Kendall's $\tau$ | p-value |
| Naive Monte Carlo | Upper | 0.151 | 0.330 | 0.108 | 0.370 |
| | Lower | 0.086 | 0.356 | 0.137 | 0.274 |
| | Mean | 0.237 | 0.228 | 0.182 | 0.352 |
| Frequentist | Upper | 0.316 | 0.476 | 0.394 | 0.426 |
| | Lower | 0.186 | 0.307 | 0.216 | 0.395 |
| | Mean | 0.146 | 0.265 | 0.413 | 0.652 |

