# OpenReview forum: "Can You Trust Your Model? Constructing Uncertainty Approximations  Guaranteeing Validity of Glioma Segmentation Explanations"
_MIDL.io/2026/Conference — MIDL 2026 Poster_

### Official Review · Reviewer_EJai · 2026-01-08

**Confidence:** 4
**Preliminary Rating:** 2
**Final Rating:** 4

**Summary:**

The authors model the uncertainty of Shapley values for explanation in glioma segmentation. They do this using a frequentist framework that is supposed to provide calibrated uncertainties with formal guarantees, similar to conformal prediction. The experiments include a comparison against a baseline that provides uncertainties based on crossfold validation.

**Strengths:**

The paper is well-written and clearly structured. Especially the methods section is very thorough and provides many details on the methods used in the experiment. The experiments also demonstrate a clear performance advantage for their proposed approach in terms of uncertainty quality.

**Weaknesses:**

Unfortunately I believe this paper is not suited for the conference due to several shortcomings. My main critique points are:
- Unclear contribution (I think especially the first point of the listings of your contributions is concerning an already existing method which makes your part rather an application as you are basically applying conformal prediction to shap values as introduced by Marx et al.)
- Sometimes a bit sloppy on the math (see details below)
- The chosen baseline is not what is usually used for uncertainty quantification methods. A more appropriate comparison would be other methods that would yield uncertainty intervals for Shap values

**Detailed Comments:**

Please find my detailed critique points here:
- Clinical motivation unclear on why it is useful to have a distribution over shap values
- in 2.1, why do you define N as a set of contrasts instead of just directly the number of contrasts?
- in 2.2, for completeness you should also constrain both lambdas to be in [0,1] if you already constrain them to sum up to 1
- in 2.5 you are basically introducing conformal prediction, why aren't you calling it like that? Also it is unclear to me how you obtain w*
- are you applying algorithm 1 on a separate calibration set? If yes, what are the details on size etc.?
- you introduced in 2.5 that you have coverage guarantees. Why aren't you evaluating coverage and instead use Kendall's tau?

**Justification Of Final Rating:**

The authors were able to clarify a substantial part of my concerns and misunderstandings and re-organized the paper such that the contribution is more clear to me now. Using correct definitions helps the reader through the paper and with the new information I do believe that the paper could be an interesting contribution.

**Justification Of The Preliminary Rating:**

Overall, I don't think that the paper can be changed easily into something that is valuable for the conference. Simply applying conformal prediction to another problem is not enough in my opinion to be a valuable contribution to the MIDL community. At least it should be clear what one can do with distributions over shap values in this clinical context. Together with a weak evaluation I suggest to reject the paper.

**Questions To Address In The Rebuttal:**

Please see the main points above for this

---

> ### Author Response · Authors · 2026-01-25
>
> * Unclear contribution
>
> Our contributions provide a clinical application for explaining tumor segmentation model selection with statistical proved clinical utility. As we demonstrated (Introduction, Figure 4), models may show high performance but produce varying explanations for decisions. Within a clinical setting, idiosyncratic feature dependencies may have serious impacts on patient outcomes. Furthermore, there is also the inundation of new AI tools that tout high performance [7]. With so many options, we see our tool as an avenue for filtering models that have true clinical utility.
> We previously showed that greater agreement between U-Net Shapley rankings and clinical rankings of MRI contrasts was correlated with higher Dice performance (Ren et al., 2025a). This led us to hypothesize that clinically informed rankings of contrasts, which directly parallels Shapley values assigned to MRI contrasts, approaches the optimal explanation for tumor segmentation and can serve as a “filter” for models. Thus, we viewed applying conformal analysis to create uncertainty intervals with statistical guarantees of the optimal explanation as the next step.
> We now show statistically significant correlation between the uncertainty intervals and our clinical consensus ranking (Table 2), validation of our belief in clinically grounded explanations as an approximation of the optimal explanation for the glioma population. By providing support for our criteria, we have a clear path for both training and selecting models that have rigorous clinical validation and can be confidently implemented within clinical settings.
>
> * The chosen baseline is not what is usually used for uncertainty quantification.
>
> We used deep ensemble, rather than Monte Carlo dropout (both are classical UQ methods). We picked deep ensemble for two primary reasons.  First, our model is designed to capture uncertainty arising from the underlying data distribution, rather than uncertainty induced solely by model stochasticity. Second, we empirically observed that standard Monte Carlo dropout yields limited variability in Shapley values when the Dice does not degrade, whereas the deep ensemble provides more informative and discriminative uncertainty estimates. In the manuscript, we changed the name of the naive approach from Monte Carlo to deep ensemble to avoid confusion.
>
> * in 2.1, why do you define N as a set of contrasts instead of just directly the number of contrasts?
>
> In our work, we define N N = {n1, n2, n3, n4} as a set, corresponding to the MRI contrasts. We adopt the set-based notation introduced in the original Shapley explanation (see Equation 4 in [6]), as it enables an explicit specification of which subsets are included or excluded in calculation We have explicitly highlighted the set operation by fixing N as a set and adding the symbol to emphasize the role of n_i (Updated Equation 2). This notation is necessary because the calculation of Shapley values requires iterating over all subsets S ⊆ N\{ni}, with the corresponding contribution given by D_ω^(r)(S ∪ {ni}) - D_ω^(r)(S).
>
> * in 2.2, for completeness you should also constrain both lambdas …?
>
> We agree that explicitly stating λ_Dice, λ_Focal ∈ [0,1] improves mathematical completeness and clarity. We completed the notation to include this explicit constraint following Equation 1.
> * in 2.5 you are basically introducing conformal prediction? how you obtain w*?
>
> We have updated the title of Section 2.5 from “Uncertainty Intervals for Shapley Values” to “Uncertainty Intervals for Shapley Values through Conformal Analysis”, which we believe will help readers better understand the topic.
> For the optimal model w*, we do not have direct access to its true value. Following the approach of Marx et al. (2023), we approximate w* using a confidence interval constructed around w^. This interval is designed such that, with significance level alpha, it captures the true optimal model w*. In other words, the confidence interval centered at w^ provides a probabilistic guarantee that w* lies within it.
>
> * are you applying algorithm 1 on …?
>
> We did not use a separate calibration set as the goal of the Shapley interval is to generate intervals for the entire dataset, rather than being limited to a validation set. Algorithm 1 is applied to the full dataset, which includes 1,351 subjects. We have added training details at the end of Section 2.5.
>
> * Why Kendall’s Tau and not evaluating coverage?
>
> Clinical rankings and Shapley values operate in fundamentally different spaces. Rankings are ordinal (1-4) while Shapley values are unbounded. Humans cannot create Shapley values; they can only rank contrasts by perceived importance. Therefore, we cannot evaluate whether clinical ranking values fall within Shapley value intervals and must use rank correlation to compare the two. We have provided a new figure (Figure 2) that illustrates our method of harmonizing these two metrics.

---

### Official Review · Reviewer_54o1 · 2026-01-09

**Confidence:** 5
**Preliminary Rating:** 4

**Summary:**

This paper proposes a frequentist framework to derive statistically valid uncertainty intervals for contrast-level Shapley values in glioma segmentation, addressing model multiplicity and the unreliability of naïve Monte-Carlo–based uncertainty. Using a U-Net trained on BraTS 2024 GoAT, the authors contrast naïve CV-based intervals with uniform-convergence–based confidence sets that approximate the Pareto frontier of models balancing loss and Shapley value. Validation is performed by comparing model-derived rankings with clinical protocol and consensus rankings, showing naïve intervals fail to align while frequentist bounds moderately correlate with human consensus, especially upper-bound and mean estimates.

**Strengths:**

Clear problem motivation: Model multiplicity and explanation uncertainty are cleanly described and clinically relevant.
Methodologically novel adaptation of Marx et al. (2023) from classification to segmentation and Shapley explanations.
Excellent experimental design:
- BraTS 2024 GoAT dataset with >1300 subjects.
- Systematic Cross-Validation pipeline with region-wise Dice and uncertainty.

Compelling comparative evidence: Frequentist intervals outperform naïve estimates (Table 2) in aligning with consensus reasoning.
Good clinical grounding: Incorporates protocol and case-level consensus labels.

**Weaknesses:**

Architecture and training are baseline: Only 2D/3D U-Net validated; no transformer models or ensembles considered.
No statistical justification for parameters like α=0.05 and εₙ=0.05 beyond citation to prior work.

Clinical evaluation limited:
Annotators are medical students, not neuroradiologists.
Consensus derived from partially low-performing cases.
Computational burden not quantified: Pareto tracing requires many retrains; wall-clock cost unknown.
Interpretability conclusions could overreach—correlation is “moderate” rather than decisive.

**Detailed Comments:**

Clarify practical feasibility: How many λ evaluations (the sequence of mixture weights), compute days? (Algorithm 1)
Report the variability of Dice across λ-optimized models to show that the loss remains clinically acceptable as Shapley values vary.
Consider per-region intervals, not just their average across regions (Equation 4 hides clinically meaningful differences).
Figure 3(b) nicely illustrates multiplicity; consider quantifying dispersion (standard deviation, entropy).
State explicitly whether lower/upper bounds always capture clinical rankings in any MRI sub-region.
Include additional discussion of why Protocol ≠ Consensus divergence emerges (Table 2).

**Justification Of The Preliminary Rating:**

This paper identifies and tackles a clinically important limitation of post-hoc explainability, uncertainty and instability of feature attribution under model multiplicity. The authors provide a well-reasoned frequentist framework and empirically demonstrate that naïve cross-validation uncertainty is insufficient to validate alignment of model reasoning with clinical knowledge, whereas statistically guaranteed intervals correlate moderately with human consensus rankings.
The contribution is incremental relative to Marx et al. (2023), but the adaptation to multi-contrast segmentation and integration with clinical workflows adds meaningful value for the MIDL community. Limitations include a narrow model family (U-Net only), limited annotator expertise, and no exploration of computational or hyperparameter sensitivity. Nevertheless, the paper is rigorous and well-organized, and it proposes a principled tool for explainability validation. I lean toward acceptance, provided that the rebuttal clarifies the generality and practical feasibility of the method.

**Questions To Address In The Rebuttal:**

How sensitive are results to \epsilon_n?
Could different uniform-convergence bounds tighten intervals and affect correlation?
Are rankings consistent across tumor sub-regions?
“Average Shapley” may mask region-specific alignment.

Would transformer or diffusion backbones show the same trend?

Could a neuroradiologist's consensus alter validation?
Medical student ranking may underestimate protocol fidelity.

---

> ### Author Response · Authors · 2026-01-25
>
> * Baseline architecture
>
> We agree additional architectures would strengthen our work. Our selection of the U-Net backbone over Transformer architectures is grounded in our prior research [5] of clinical interpretability. We previously found that Swin-UNETR exhibits evenly distributed Shapley values across contrasts and fails to replicate the prioritization characteristic of established clinical protocols. We hypothesize that this behavior stems from architectural inductive biases, where transformers aggregate global context through token mixing, thereby homogenizing contrast contributions. However, the Convolution-based U-Net demonstrates feature attributions that are more clinically aligned, effectively capturing the specific contrast dependencies used by radiologists. Therefore, we selected the U-Net as the backbone for this initial validity framework. A systematic comparison across additional transformer and diffusion-based backbones is an important direction for future work.
>
> * Justification for parameters like α and εₙ
>
> We have added clarification in Section 2.5. α=0.05 follows standard practice for 95% confidence intervals. ε=0.05 corresponds to 2ε=0.10 tolerance on empirical loss (Equation 6). Our best model achieves 87% Dice, so this tolerance corresponds to minimum 77% Dice, capturing human-expert annotation accuracy and top-performing tumor segmentation models [1][2][3]. This ensures our confidence set explores reasonable Shapley value explanations while maintaining statistical guarantees.
>
> * Annotators are medical students
>
> We fully acknowledge this limitation and would include neuroradiologist rankings if feasible. Nevertheless, the rankings were performed by medical students with prior radiology research experience, who adhered strictly to the established clinical protocol.
>
> * Consensus derived from partially low-performing cases.
>
> This was intentional to verify medical student rankings remained stable for "difficult" cases, as protocol [4] provides global criteria. Rankers were blinded to model performance, ensuring unbiased ranking. Section 2.3 now explains this selection.
>
> * Computational burden not quantified
>
> Computational cost details added to Section 2.5.
>
> * Interpretability conclusions could overreach
>
> We agree results are moderate (τ=0.467, p=0.015 is significant but not strong) and revised language accordingly.
>
> * Clarify practical feasibility: How many λ evaluations, compute days?
>
> See #5 and updated Section 2.5 for more details.
>
> * Report the variability of Dice across λ-optimized models
>
> We have added more detailed explanations of the lambda value used in the calculation.
>
> * Consider per-region intervals
>
> This is a fair point to bring up and we considered evaluating per-region intervals in our methodology. However, we realized that the uncertainty intervals had to align with the clinical rankings derived from the protocol, as overall tumor contrast rankings. Otherwise, it would be very useful to observe interval ranking alignment by region.
>
> * Consider quantifying dispersion (standard deviation, entropy).
>
> We have added standard deviation, entropy in Figure 4 (previously 3) to better explain the results.
>
> * State explicitly whether lower/upper bounds always capture clinical rankings in any MRI sub-region.
>
> We understand that the adaptation of contrast Shapley values to contrast rankings may not have been clearly communicated. Clinical rankings and Shapley values operate in different spaces. Rankings are ordinal (1-4) while Shapley values are continuous contributions to segmentation performance. Humans cannot directly assess Shapley values; they can only rank contrasts by perceived importance. Therefore, we cannot evaluate whether clinical ranking values (e.g., "rank 2") fall within Shapley value intervals (e.g., [0.25, 0.35]). This is why we use Kendall's τ rank correlation (Table 2), which compares the ordering of contrasts rather than raw values. We clarified this decision in the revised manuscript and provided a new figure (Figure 2) that illustrates our method of harmonizing these two metrics.
>
>
> * Include additional discussion of why Protocol ≠ Consensus divergence emerges
>
> We have added more discussion in Section 4.
>
> * How sensitive are results to \epsilon? Could different ...?
>
> Please see Weakness #2. The parameter ε selected in the paper is clinically justified. We have added a sensitivity analysis in the appendix to examine the effect of choosing a larger ε. The results show that under a large ε, there is no correlation between the Shapley interval rankings and either the consensus or the protocol rankings.
>
> * Could a neuroradiologist's consensus alter validation?
>
> The impact is uncertain. If neuroradiologists show stronger protocol adherence, correlation could actually weaken. However, neuroradiologists likely do not replicate protocol rigidly and would provide more refined case-specific judgment with lower inter-rater variability than medical students.

---

### Official Review · Reviewer_Xvpg · 2026-01-10

**Confidence:** 4
**Preliminary Rating:** 3
**Final Rating:** 4

**Summary:**

In this paper, the authors develop an elegant approach to construct statistically valid uncertainty intervals for contrast-level shapely values in Glioma segmentation. Their frequentist uncertainty quantification framework provides a guarantee that optimal (clinically expected) explanation lies within its confidence interval.

**Strengths:**

The paper introduces a mathematically well-founded frequentist framework that constructs statistically valid confidence intervals for contrast-level Shapley values, guaranteeing coverage of the “optimal” model explanation rather than just a single trained network.

It further strengthens its contribution by empirically contrasting these intervals with naive cross-validation based estimates and evaluating their alignment with consensus clinical rankings on a large BraTS 2024 GoAT glioma segmentation cohort.

**Weaknesses:**

1. Details of the U-Net architecture used are missing.
2. For providing a reasonable explanation, is there a trade-off with model accuracy? In a regular segmentation approach, the 4 input MRI contrasts are provided as separate input channels to a segmentation network, while the network computes latent space features that are formed from a combination of these channels. This interaction is missing in the proposed approach (as seen in Equation 1, and the explanation of $L_{Dice}$ thereof). Since this is necessary for the proposed explanation strategy, it remains unclear if the explanation comes at the cost of accuracy.
3. In Section 2.4, it appears that the Shapely values from both training and validation sets are used for ranking. Is the use of training dataset justified to compute Shapely rankings, where the accuracy would be artificially high and thus the explanation would also align more with the clinical expectation?
4. It is unclear (in Section 2.4) why the actual Shapely values are being ignored while computing the mean rankings (since only the ranks of different contrasts are being considered). Can the Shapely values be used as weights for computing the mean ranking? It gives an impression that the comparison with the Naive approach is not very fair.

**Detailed Comments:**

Please see the section on weaknesses.

**Justification Of Final Rating:**

The authors have adequately addressed my main concerns in their rebuttal and in the revised manuscript. I am now more confident in the robustness and relevance of the work and view it suitable for acceptance.

**Justification Of The Preliminary Rating:**

Providing explanations for medical image segmentation and quantifying uncertainties in them are crucial for improving trustworthiness of the automatic segmentations. Although the paper proposes a mathematically sound and clinically relevant uncertainty framework, key methodological choices around the segmentation architecture, potential accuracy-explanation trade-offs, Shapley computation on training data, and the use of ranks rather than Shapley magnitudes are insufficiently justified. These gaps limit my confidence in the robustness and fairness of the empirical evidence, so I consider a borderline recommendation appropriate.

**Questions To Address In The Rebuttal:**

Please see the section on weaknesses.

---

> ### Author Response · Authors · 2026-01-25
>
> * Details of the U-Net architecture used are missing.
>
> Thank you for pointing out the lack of architectural details. We use a standard multi-channel 3D U-Net implemented in the MONAI library, with four encoder–decoder stages and skip connections, taking four MRI contrasts as input. We have added a detailed architectural description to the Methods section for clarity.
>
> * For providing a reasonable explanation, is there a trade-off with model accuracy? In a regular segmentation approach, the 4 input MRI contrasts are provided as separate input channels to a segmentation network, while the network computes latent space features that are formed from a combination of these channels. This interaction is missing in the proposed approach (as seen in Equation 1, and the explanation of $L_{Dice}$ thereof). Since this is necessary for the proposed explanation strategy, it remains unclear if the explanation comes at the cost of accuracy.
>
> We agree that in multi-contrast MRI segmentation with U-Net, the network learns latent representations that strongly couple information across contrasts. The proposed method in the paper does not represent a trade-off between explanation and accuracy. The observed decrease in Dice score when a contrast is removed is not a limitation of the model, but the fundamental mechanism by which Shapley values quantify the model’s functional dependence on each contrast.
> In detail, the original Shapley value framework neither assumes independence between contrasts nor ignores such interactions. Instead, Shapley-based explanations explicitly account for interactions by averaging the marginal contribution of each contrast over all possible subsets of the remaining contrasts. As a result, each contrast is evaluated in both the presence and absence of others, inherently capturing higher-order interactions in the learned representation.
>
> * In Section 2.4, it appears that the Shapely values from both training and validation sets are used for ranking. Is the use of training dataset justified to compute Shapely rankings, where the accuracy would be artificially high and thus the explanation would also align more with the clinical expectation?
>
> Our naive Monte Carlo uncertainty intervals (Section 2.4) indeed pool Shapley values from all 1,351 subjects across all folds, including training subjects. While this may inflate Shapley values and artificially improve apparent clinical alignment, we note that even with this potential inflation, naive intervals still failed to show significant correlation with clinical rankings (Figure 2). This, in a way, strengthens the argument. If naive intervals cannot align with clinical reasoning even when potentially inflated by training-set overfitting, they are fundamentally unreliable for explanation validation.
>
> * It is unclear (in Section 2.4) why the actual Shapely values are being ignored while computing the mean rankings (since only the ranks of different contrasts are being considered). Can the Shapely values be used as weights for computing the mean ranking? It gives an impression that the comparison with the Naive approach is not very fair.
>
> Thank you for this comment. We believe you are asking why the Naive approach utilizes Shapley rankings and not values. There are actually two “naive” approaches in the paper, the first being found in 2.4 that allows direct comparison of model Shapley rankings (converted from values) and the clinical rankings. This serves as a quick initial comparison of model and clinical rankings and highlights the initial mismatch. We then compare the model Shapley value intervals to the clinical rankings by comparing rank correlation, found in 2.5: Naive Monte Carlo Comparison. This naive comparison serves as the true frame of reference for the comparison between the frequentist uncertainty intervals and the clinical rankings, again assessed through rank correlation (2.5: Frequentist Uncertainty Intervals). We have adjusted language in the paper to clarify that the first naive approach (2.4) is not the main method for comparison to the frequentist approach.

---

### Author Rebuttal · Authors · 2026-01-25

**Rebuttal:**

We sincerely thank the reviewers for their thoughtful and constructive feedback. In this rebuttal, we focus on revisions regarding architectural choices, statistical justification, clinical validity, computational feasibility, hyperparameter selection, and the scope of our interpretability claims. A revised PDF is provided with all changes highlighted in red. The following references are used in our responses to the reviewers.

References

[1] Pemberton et al., *Multi-class glioma segmentation on real-world data with missing MRI sequences: comparison of three deep learning algorithms.* Scientific Reports 13, 18911 (2023).

[2] Tillmanns et al., *Identifying clinically applicable machine learning algorithms for glioma segmentation: recent advances and discoveries.* Neuro-Oncology Advances 4(1), vdac093 (2022).

[3] Masafumi et al., *Evaluation of the image quality index with MRI motion artifacts on tumor segmentation using deep learning* (2025).

[4] Ujjwal et al., *The RSNA-ASNR-MICCAI BraTS 2021 benchmark on brain tumor segmentation and radiogenomic classification.* (2021).

[5] Ren et al., *Here Comes the Explanation: A Shapley Perspective on Multi-contrast Medical Image Segmentation.* (2025).

[6] Lundberg, S. M., and Lee, S.-I., *A unified approach to interpreting model predictions.* Advances in Neural Information Processing Systems 30 (2017).

[7] Brady, A. P. et al., *Developing, purchasing, implementing and monitoring AI tools in radiology: practical considerations.* Canadian Association of Radiologists Journal 75(2), 226–244 (2024).

**Supporting Material:**

/attachment/473509192d9318cd8ea1518ba0d036113cfe9050.pdf

---

### Meta-Review · Area_Chair_ptjT · 2026-02-01

**Recommendation:** Accept (Poster)
**Confidence:** 3

**Metareview:**

After reading the reviewers' comments and the responses, the reviewers' comments are fair and make sense with details. The rebuttal has done a good job. Even with a weak reject, I think the paper can be accepted as a poster.

---

### Decision · Program_Chairs · 2026-02-13

Accept (Poster)